# Distributed Weight Consolidation: A Brain Segmentation Case Study

**Patrick McClure**
National Institute of Mental Health
patrick.mcclure@nih.gov

**Charles Y. Zheng**
National Institute of Mental Health
charles.zheng@nih.gov

**Jakub R. Kaczmarzyk**
Massachusetts Institute of Technology
jakubk@mit.edu

**John A. Lee**
National Institute of Mental Health
john.rodgers-lee@nih.gov

**Satrajit S. Ghosh**
Massachusetts Institute of Technology
satra@mit.edu

**Dylan Nielson**
National Institute of Mental Health
dylann.nielson@nih.gov

**Peter Bandettini**
National Institute of Mental Health
bandettini@nih.gov

**Francisco Pereira**
National Institute of Mental Health
francisco.pereira@nih.gov

## Abstract

Collecting the large datasets needed to train deep neural networks can be very difficult, particularly for the many applications for which sharing and pooling data is complicated by practical, ethical, or legal concerns. However, it may be the case that derivative datasets or predictive models developed within individual sites can be shared and combined with fewer restrictions. Training on distributed data and combining the resulting networks is often viewed as continual learning, but these methods require networks to be trained sequentially. In this paper, we introduce distributed weight consolidation (DWC), a continual learning method to consolidate the weights of separate neural networks, each trained on an independent dataset. We evaluated DWC with a brain segmentation case study, where we consolidated dilated convolutional neural networks trained on independent structural magnetic resonance imaging (sMRI) datasets from different sites. We found that DWC led to increased performance on test sets from the different sites, while maintaining generalization performance for a very large and completely independent multi-site dataset, compared to an ensemble baseline.

## 1   Introduction

Deep learning methods require large datasets to perform well. Collecting such datasets can be very difficult, particularly for the many applications for which sharing and pooling data is complicated by practical, ethical, or legal concerns. One prominent application is human subjects research, in which researchers may be prevented from sharing data due to privacy concerns or other ethical considerations. These concerns can significantly limit the purposes for which the collected data can be used, even within a particular collection site. If the datasets are collected in a clinical setting, they may be subject to many additional constraints. However, it may be the case that derivative

datasets or predictive models developed within individual sites can be shared and combined with fewer restrictions.

In the neuroimaging literature, several platforms have been introduced for combining models trained on different datasets, such as ENIGMA ([29], for meta-analyses) and COINSTAC ([23], for distributed training of models). Both platforms support combining separately trained models by averaging the learned parameters. This works for convex methods (e.g. linear regression), but does not generally work for non-convex methods (e.g. deep neural networks, DNNs). [23] also discussed using synchronous stochastic gradient descent training using server-client communication; this assumes that all of the training data is simultaneously available. Also, for large models such as DNNs, the bandwidth required could be problematic, given the need to transmit gradients at every update.

Learning from non-centralized datasets using DNNs is often viewed as continual learning, a sequential process where a given predictive model is updated to perform well on new datasets, while retaining the ability to predict on those previously used for training [32, 4, 21, 16, 18]. Continual learning is particularly applicable to problems with shifting input distributions, where the data collected in the past may not represent data collected now or in the future. This is true for neuroimaging, since the statistics of MRIs may change due to scanner upgrades, new reconstruction algorithms, different sequences, etc. The scenario we envisage is a more complex situation where multiple continual learning processes may take place non-sequentially. For instance, a given organization produces a starting DNN, which different, independent sites will then use with their own data. The sites will then contribute back updated DNNs, which the organization will use to improve the main DNN being shared, with the goal of continuing the sharing and consolidation cycle.

Our application is segmentation of structural magnetic resonance imaging (sMRI) volumes. These segmentations are often generated using the Freesurfer package [8], a process that can take close to a day for each subject. The computational resources for doing this at a scale of hundreds to thousands of subjects are beyond the capabilities of most sites. We use deep neural networks to predict the Freesurfer segmentation *directly* from the structural volumes, as done previously by other groups [26, 27, 7, 6]. We train several of those networks – each using data from a different site – and then consolidate their weights. We show that this results in a model with improved generalization performance in test data from these sites, as well as a very large, completely independent multi-site dataset.

## 2 Data and Methods

### 2.1 Datasets

We use several sMRI datasets collected at different sites. We train networks using 956 sMRI volumes collected by the Human Connectomme Project (HCP) [30], 1,136 sMRI volumes collected by the Nathan Kline Institute (NKI) [22], 183 sMRI volumes collected by the Buckner Laboratory [2], and 120 sMRI volumes from the Washington University 120 (WU120) dataset [24]. In order to provide an independent estimate of how well a given network generalizes to any new site, we also test networks on a completely held-out dataset consisting of 893 sMRI volumes collected across several institutions by the ABIDE project [5].

### 2.2 Architecture

Several deep neural network architectures have been proposed for brain segmentation, such as U-net [26], QuickNAT [27], HighResNet [18] and MeshNet [7, 6]. We chose MeshNet because of its relatively simple structure, its lower number of learned parameters, and its competitive performance.

MeshNet uses dilated convolutional layers [31] due to the 3D structural nature of sMRI data. The output of these discrete volumetric dilated convolutional layers can be expressed as:

$$(\mathbf{w}_f *_l \mathbf{h})_{i,j,k} = \sum_{\tilde{i}=-a}^{a} \sum_{\tilde{j}=-b}^{b} \sum_{\tilde{k}=-c}^{c} w_{f,\tilde{i},\tilde{j},\tilde{k}} h_{i-l\tilde{i},j-l\tilde{j},k-l\tilde{k}} = (\mathbf{w}_f *_l \mathbf{h})_{\mathbf{v}} = \sum_{\mathbf{t} \in \mathcal{W}_{abc}} w_{f,\mathbf{t}} h_{\mathbf{v}-l\mathbf{t}}. \quad (1)$$

| Layer | Filter | Pad | Dilation ($l$) | Function | Layer | Filter | Pad | Dilation ($l$) | Function |
|---|---|---|---|---|---|---|---|---|---|
| 1 | 96x3$^3$ | 1 | 1 | ReLU | 5 | 96x3$^3$ | 4 | 4 | ReLU |
| 2 | 96x3$^3$ | 1 | 1 | ReLU | 6 | 96x3$^3$ | 8 | 8 | ReLU |
| 3 | 96x3$^3$ | 1 | 1 | ReLU | 7 | 96x3$^3$ | 1 | 1 | ReLU |
| 4 | 96x3$^3$ | 2 | 2 | ReLU | 8 | 50x1$^3$ | 0 | 1 | Softmax |

Table 1: The Meshnet-like dilated convolutional neural network architecture for brain segmentation.

where $h$ is the input to the layer, $a$, $b$, and $c$ are the bounds for the $i$, $j$, and $k$ axes of the filter with weights $\mathbf{w}_f$, $(i, j, k)$ is the voxel, $\mathbf{v}$, where the convolution is computed. The set of indices for the elements of $\mathbf{w}_f$ can be defined as $\mathcal{W}_{abc} = \{-a, ..., a\} \times \{-b, ..., b\} \times \{-c, ..., c\}$. The dilation factor $l$ allows the convolution kernel to operate on every $l$-th voxel, since adjacent voxels are expected to be highly correlated. The dilation factor, number of filters, and other details of the MeshNet-like architecture that we used for all experiments is shown in Table 1.

## 2.3 Bayesian Inference in Neural Networks

### 2.3.1 Maximum a Posteriori Estimate

When training a neural network, the weights of the network, $\mathbf{w}$ are learned by optimizing $\text{argmax}_{\mathbf{w}} p(\mathbf{w}|\mathcal{D})$ where $\mathcal{D} = \{(\mathbf{x}_1, \mathbf{y}_1), ..., (\mathbf{x}_N, \mathbf{y}_N)\}$ and $(\mathbf{x}_n, \mathbf{y}_n)$ is the $n$th input-output example, per maximum likelihood estimation (MLE). However, this often overfits, so we used a prior on the network weights, $p(\mathbf{w})$, to obtain a maximum a posteriori (MAP) estimate, by maximizing:

$$\mathcal{L}_{MAP}(\mathbf{w}) = \sum_{n=1}^{N} \log p(\mathbf{y}_n|\mathbf{x}_n, \mathbf{w}) + \log p(\mathbf{w}). \tag{2}$$

### 2.3.2 Approximate Bayesian Inference

In Bayesian inference for neural networks, a distribution of possible weights is learned instead of just a MAP point estimate. Using Bayes' rule, $p(\mathbf{w}|\mathcal{D}) = p(\mathcal{D}|\mathbf{w})p(\mathbf{w})/p(\mathcal{D})$, where $p(\mathbf{w})$ is the prior over weights. However, directly computing the posterior, $p(\mathbf{w}|\mathcal{D})$, is often intractable, particularly for DNNs. As a result, an approximate inference method must be used.

One of the most popular approximate inference methods for neural networks is variational inference, since it scales well to large DNNs. In variational inference, the posterior distribution $p(\mathbf{w}|\mathcal{D})$ is approximated by a learned variational distribution of weights $q_\theta(\mathbf{w})$, with learnable parameters $\theta$. This approximation is enforced by minimizing the Kullback-Leibler divergence (KL) between $q_\theta(\mathbf{w})$, and the true posterior, $p(\mathbf{w}|\mathcal{D})$, $\text{KL}[q_\theta(\mathbf{w})||p(\mathbf{w}|\mathcal{D})]$. This is equivalent to maximizing the variational lower bound [11, 10, 3, 14, 9, 20, 19], also known as the evidence lower bound (ELBO),

$$\mathcal{L}_{ELBO}(\theta) = \mathcal{L}_{\mathcal{D}}(\theta) - \mathcal{L}_{KL}(\theta), \tag{3}$$

where $\mathcal{L}_{\mathcal{D}}(\theta)$ is

$$\mathcal{L}_{\mathcal{D}}(\theta) = \sum_{n=1}^{N} \mathbb{E}_{q_\theta(\mathbf{w})}[\log p(\mathbf{y}_n|\mathbf{x}_n, \mathbf{w})] \tag{4}$$

and $\mathcal{L}_{KL}(\theta)$ is the KL divergence between the variational distribution of weights and the prior,

$$\mathcal{L}_{KL}(\theta) = \text{KL}[q_\theta(\mathbf{w})||p(\mathbf{w})]. \tag{5}$$

Maximizing $L_{\mathcal{D}}$ seeks to learn a $q_\theta(\mathbf{w})$ that explains the training data, while minimizing $L_{KL}$ (i.e. keeping $q_\theta(\mathbf{w})$ close to $p(\mathbf{w})$) prevents learning a $q_\theta(\mathbf{w})$ that overfits to the training data.

### 2.3.3 Stochastic Variational Bayes

Optimizing Eq. 3 for deep neural networks is usually impractical to compute, due to both: (1) being a full-batch approach and (2) integrating over $q_\theta(\mathbf{w})$. (1) is often dealt with by using stochastic mini-batch optimization [25] and (2) is often approximated using Monte Carlo sampling. [14] applied these methods to variational inference in deep neural networks. They used the "reparameterization trick" [15], which formulates $q_\theta(\mathbf{w})$ as a deterministic differentiable function $\mathbf{w} = f(\theta, \epsilon)$ where $\epsilon \sim \mathcal{N}(0, I)$, to calculate an unbiased estimate of $\nabla_\theta \mathcal{L}_\mathcal{D}$ for a mini-batch, $\{(\mathbf{x}_1, \mathbf{y}_1), ..., (\mathbf{x}_M, \mathbf{y}_M)\}$, and one weight noise sample, $\epsilon_m$, for each mini-batch example:

$$\mathcal{L}_{ELBO}(\theta) \approx \mathcal{L}_\mathcal{D}^{SGVB}(\theta) - \mathcal{L}_{KL}(\theta), \tag{6}$$

where

$$\mathcal{L}_\mathcal{D}(\theta) \approx \mathcal{L}_\mathcal{D}^{SGVB}(\theta) = \frac{N}{M} \sum_{m=1}^M \log p(\mathbf{y}_m|\mathbf{x}_m, f(\theta, \epsilon_m)). \tag{7}$$

### 2.3.4 Variational Continual Learning

In Bayesian neural networks, $p(\mathbf{w})$ is often set to a multivariate Gaussian with diagonal covariance $\mathcal{N}(\mathbf{0}, \sigma_{prior}^2 I)$. (A variational distribution of the same form is called a fully factorized Gaussian (FFG).) However, instead of using a naïve prior, the parameters of a previously trained DNN can be used. Several methods, such elastic weight consolidation [16] and synaptic intelligence [32], have explored this approach. Recently, these methods have been reinterpreted from a Bayesian perspective [21, 17]. In variational continual learning (VCL) [21] and Bayesian incremental learning [17], the DNNs trained on previously obtained data, $\mathcal{D}_1$-$\mathcal{D}_{T-1}$, are used to regularize the training of a new neural network trained on $\mathcal{D}_T$ per:

$$p(\mathbf{w}|\mathcal{D}_{1:T}) = \frac{p(\mathcal{D}_{1:T}|\mathbf{w})p(\mathbf{w})}{p(\mathcal{D}_{1:T})} = \frac{p(\mathcal{D}_{1:T-1}|\mathbf{w})p(\mathcal{D}_T|\mathbf{w})p(\mathbf{w})}{p(\mathcal{D}_{1:T-1})p(\mathcal{D}_T)} = \frac{p(\mathbf{w}|\mathcal{D}_{1:T-1})p(\mathcal{D}_T|\mathbf{w})}{p(\mathcal{D}_T)}, \tag{8}$$

where $p(\mathbf{w}|\mathcal{D}_{1:T-1})$ is the network resulting from training on a sequence of datasets $\mathcal{D}_1$-$\mathcal{D}_{T-1}$.

For DNNs, computing $p(\mathbf{w}|\mathcal{D}_{1:T})$ directly can be intractable, so variational inference is iteratively used to learn an approximation, $q_\theta^T(\mathbf{w})$, by minimizing $\mathrm{KL}[q_\theta^\tau(\mathbf{w})||p(\mathbf{w}|\mathcal{D}_{1:\tau})]$ for each sequential dataset $\mathcal{D}_\tau$, with $\tau$ ranging over integers from 1 to $T$.

The sequential nature of this approach is a limitation in our setting. In many cases it is not feasible for one site to wait for another site to complete training, which can take days, in order to begin their own training.

## 2.4 Distributed Weight Consolidation

The main motivation of our method – distributed weight consolidation (DWC) – is to make it possible to train neural networks on different, distributed datasets, independently, and consolidate their weights into a single network.

### 2.4.1 Bayesian Continual Learning for Distributed Data

In DWC, we seek to consolidate several distributed DNNs trained on $S$ separate, distributed datasets, $\mathcal{D}_T = \{\mathcal{D}_T^1, ..., \mathcal{D}_T^S\}$, so that the resulting DNN can then be used to inform the training of a DNN on $\mathcal{D}_{T+1}$. The training on each dataset starts from an existing network $p(\mathbf{w}|\mathcal{D}_{1:T-1})$.

Assuming that the $S$ datasets are independent allows Eq. 8 to be rewritten as:

$$p(\mathbf{w}|\mathcal{D}_{1:T}) = \frac{p(\mathbf{w}|\mathcal{D}_{1:T-1}) \prod_{s=1}^S p(\mathcal{D}_T^s|\mathbf{w})}{\prod_{s=1}^S p(\mathcal{D}_T^s)}. \tag{9}$$

However, training one of the $S$ networks using VCL produces an approximation for $p(\mathbf{w}|\mathcal{D}_{1:T-1}, \mathcal{D}_T^s)$. Eq. 9 can be written in terms of the learned distributions, since $p(\mathbf{w}|\mathcal{D}_{1:T-1}, \mathcal{D}_T^s) = p(\mathbf{w}|\mathcal{D}_{1:T-1})p(\mathcal{D}_T^s|\mathbf{w})/p(\mathcal{D}_T^s)$ per Eq. 8:

$$p(\mathbf{w}|\mathcal{D}_{1:T}) = \frac{1}{p(\mathbf{w}|\mathcal{D}_{1:T-1})^{S-1}} \prod_{s=1}^{S} p(\mathbf{w}|\mathcal{D}_{1:T-1}, D_T^s). \tag{10}$$

$p(\mathbf{w}|\mathcal{D}_{1:T-1})$ and each $p(\mathbf{w}|\mathcal{D}_{1:T-1}, \mathcal{D}_T^s)$ can be learned and then used to compute $p(\mathbf{w}|\mathcal{D}_{1:T})$. This distribution can then be used to learn $p(\mathbf{w}|\mathcal{D}_{1:T+1})$ per Eq. 8.

### 2.4.2 Variational Approximation

In DNNs, however, directly calculating these probability distributions can be intractable, so variational inference is used to learn an approximation, $q_\theta^{T,s}(\mathbf{w})$, for $p(\mathbf{w}|\mathcal{D}_{1:T-1}, \mathcal{D}_T^s)$ by minimizing $\mathrm{KL}[q_\theta^T(\mathbf{w})||p(\mathbf{w}|\mathcal{D}_{1:T-1}, \mathcal{D}_T^s)]$. This results in approximating Eq. 10 using:

$$p(\mathbf{w}|\mathcal{D}_{1:T}) \approx \frac{1}{q_\theta^{T-1}(\mathbf{w})^{S-1}} \prod_{s=1}^{S} q_\theta^{T,s}(\mathbf{w}). \tag{11}$$

### 2.4.3 Dilated Convolutions with Fully Factorized Gaussian Filters

Although more complicated variational families have recently been explored in DNNs, the relatively simple FFG variational distribution can do as well as, or better than, more complex methods for continual learning [17]. In this paper, we use dilated convolutions with FFG filters. This assumes each of the $F$ filters are independent (i.e. $p(\mathbf{w}) = \prod_{f=1}^{F} p(\mathbf{w}_f)$), that each weight within a filter is also independent (i.e. $p(\mathbf{w}_f) = \prod_{\mathbf{t} \in \mathcal{W}_{abc}} p(w_{f,\mathbf{t}})$), and that each weight is Gaussian (i.e. $w_{f,\mathbf{t}} \sim \mathcal{N}(\mu_{f,\mathbf{t}}, \sigma_{f,\mathbf{t}}^2)$) with learnable parameters $\mu_{f,\mathbf{t}}$ and $\sigma_{f,\mathbf{t}}$. However, as discussed in [14, 20], randomly sampling each weight for each mini-batch example can be computationally expensive, so the fact that the sum of independent Gaussian variables is also Gaussian is used to move the noise from the weights to the convolution operation. For, dilated convolutions, this is described by

$$(\mathbf{w}_f *_l \mathbf{h})_{\mathbf{v}} \sim \mathcal{N}(\mu_{f,\mathbf{v}}^*, (\sigma_{f,\mathbf{v}}^*)^2), \tag{12}$$

where

$$\mu_{f,\mathbf{v}}^* = \sum_{\mathbf{t} \in \mathcal{W}_{abc}} \mu_{f,\mathbf{t}} h_{\mathbf{v}-l\mathbf{t}} \tag{13}$$

and

$$(\sigma_{f,\mathbf{v}}^*)^2 = \sum_{\mathbf{t} \in \mathcal{W}_{abc}} \sigma_{f,\mathbf{t}}^2 h_{\mathbf{v}-l\mathbf{t}}^2. \tag{14}$$

Eq. 12 can be rewritten using the Gaussian "reparameterization trick":

$$(\mathbf{w}_f *_l \mathbf{h})_{\mathbf{v}} = \mu_{f,\mathbf{v}}^* + \sigma_{f,\mathbf{v}}^* \epsilon_{f,\mathbf{v}} \text{ where } \epsilon_{f,\mathbf{v}} \sim \mathcal{N}(0, 1). \tag{15}$$

### 2.4.4 Consolidating an Ensemble of Fully Factorized Gaussian Networks

Eq. 11 can be used to consolidate an ensemble of distributed networks in order to allow for training on new datasets. Eq. 11 can be directly calculated if $q_\theta^{T-1}(w_{f,\mathbf{t}}) = \mathcal{N}(\mu_{f,\mathbf{t}}^0, (\sigma_{f,\mathbf{t}}^0)^2)$ and $q_\theta^{T,s}(w_{f,\mathbf{t}}) = \mathcal{N}(\mu_{f,\mathbf{t}}^s, (\sigma_{f,\mathbf{t}}^s)^2)$ are known, resulting in $p(\mathbf{w}|\mathcal{D}_{1:T})$ also being an FFG per

$$p(w_{f,\mathbf{t}}|\mathcal{D}_{1:T}) \stackrel{\propto}{\sim} e^{(S-1)\frac{(w_{f,0,\mathbf{t}}-\mu_{f,\mathbf{t}}^0)^2}{2(\sigma_{f,\mathbf{t}}^0)^2}} \prod_{s=1}^{S} e^{\frac{-(w_{f,s,\mathbf{t}}-\mu_{f,\mathbf{t}}^s)^2}{2(\sigma_{f,\mathbf{t}}^s)^2}} \qquad (16)$$

and

$$p(w_{f,\mathbf{t}}|\mathcal{D}_{1:T}) \approx \mathcal{N}\left( \frac{\sum_{s=1}^{S} \frac{\mu_{f,\mathbf{t}}^s}{(\sigma_{f,\mathbf{t}}^s)^2} - \sum_{s=1}^{S-1} \frac{\mu_{f,\mathbf{t}}^0}{(\sigma_{f,\mathbf{t}}^0)^2}}{\sum_{s=1}^{S} \frac{1}{(\sigma_{f,\mathbf{t}}^s)^2} - \sum_{s=1}^{S-1} \frac{1}{(\sigma_{f,\mathbf{t}}^0)^2}}, \frac{1}{\sum_{s=1}^{S} \frac{1}{(\sigma_{f,\mathbf{t}}^s)^2} - \sum_{s=1}^{S-1} \frac{1}{(\sigma_{f,\mathbf{t}}^0)^2}} \right). \qquad (17)$$

Eq. 17 follows from Eq. 16 by completing the square inside the exponent and matching the parameters to the multivariate Gaussian density; it is defined when $\sum_{s=1}^{S} \frac{1}{(\sigma_{f,\mathbf{t}}^s)^2} - \sum_{s=1}^{S-1} \frac{1}{(\sigma_{f,\mathbf{t}}^0)^2} > 0$. To ensure this, we constrained $(\sigma_{f,\mathbf{t}}^0)^2 \geq (\sigma_{f,\mathbf{t}}^s)^2$. This should be the case if the loss is optimized, since $\mathcal{L}_\mathcal{D}$ should pull $(\sigma_{f,\mathbf{t}}^s)^2$ to 0 and $\mathcal{L}_{KL}$ pulls $(\sigma_{f,\mathbf{t}}^s)^2$ towards $(\sigma_{f,\mathbf{t}}^0)^2$. $p(w_{f,\mathbf{t}}|\mathcal{D}_{1:T})$ can then be used as a prior for training another variational DNN.

## 3 Experiments

### 3.1 Experimental Setup

#### 3.1.1 Data Preprocessing

The only pre-processing that we performed was conforming the input sMRIs with Freesurfer's mri_convert, which resized all of the volumes used to 256x256x256 1 mm voxels . We computed 50-class Freesurfer [8] segmentations, as in [6], for all subjects in each of the datasets described earlier. These were used as the labels for prediction. A 90-10 training-test split was used for the HCP, NKI, Buckner, and WU120 datasets. During training and testing, input volumes were individually z-scored across voxels. We split each input volume into 512 non-overlapping 32x32x32 sub-volumes, as in [7, 6].

#### 3.1.2 Training Procedure

All networks were trained with Adam [13] and an initial learning rate of 0.001. The MAP networks were trained until convergence. The subsequent networks were trained until the training loss started to oscillate around a stable loss value. These networks trained much faster than the MAP networks, since they were initialized with previously trained networks. Specifically, we found that using VCL

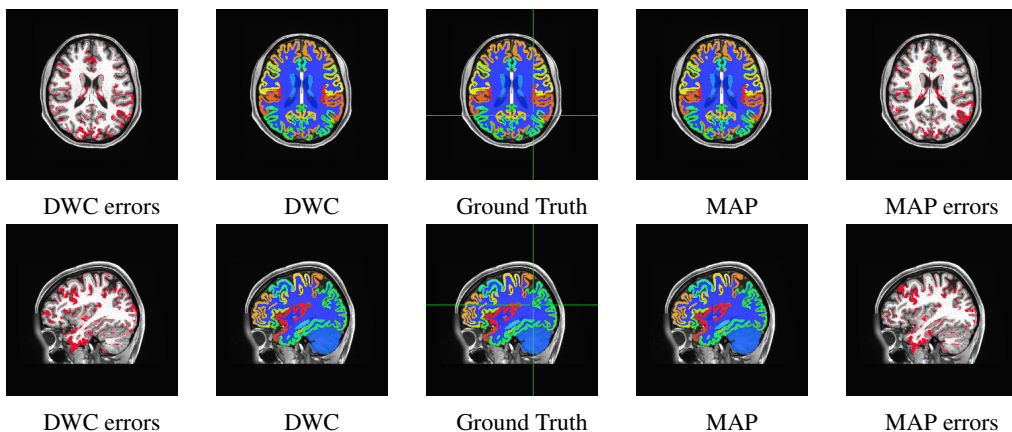

|       |     |              |     |            |
| DWC errors | DWC | Ground Truth | MAP | MAP errors |

|       |     |              |     |            |
| DWC errors | DWC | Ground Truth | MAP | MAP errors |

Figure 1: The axial and sagittal segmentations produced by DWC and the $HNBW_{MAP}$ baseline on a HCP subject. The subject was selected by matching the subject specific Dice with the average Dice across HCP. Segmentations errors for all classes are shown in red in the respective plot.

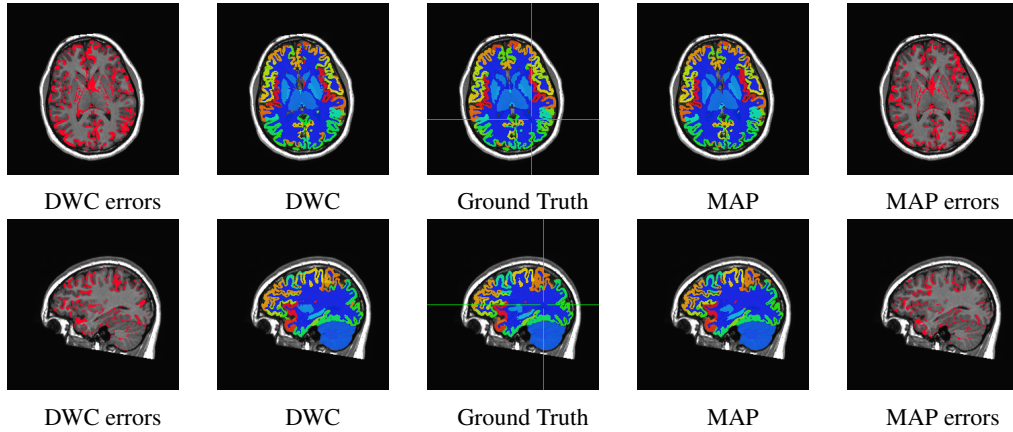

Figure 2: The axial and sagittal segmentations produced by DWC and the $HNBW_{MAP}$ baseline on a NKI subject. The subject was selected by matching the subject specific Dice with the average Dice across NKI. Segmentations errors for all classes are shown in red in the respective plot.

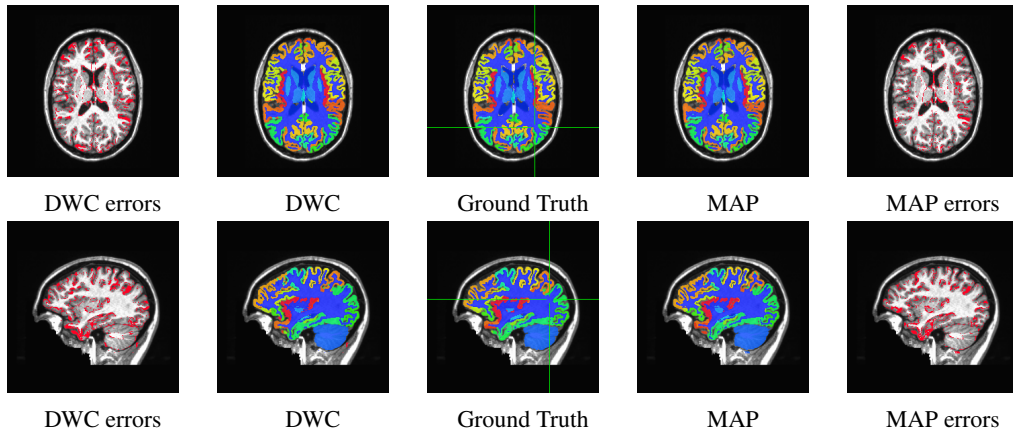

Figure 3: The axial and sagittal segmentations produced by DWC and the $HNBW_{MAP}$ baseline on a Buckner subject. The subject was selected by matching the subject specific Dice with the average Dice across Buckner. Segmentations errors for all classes are shown in red in the respective plot.

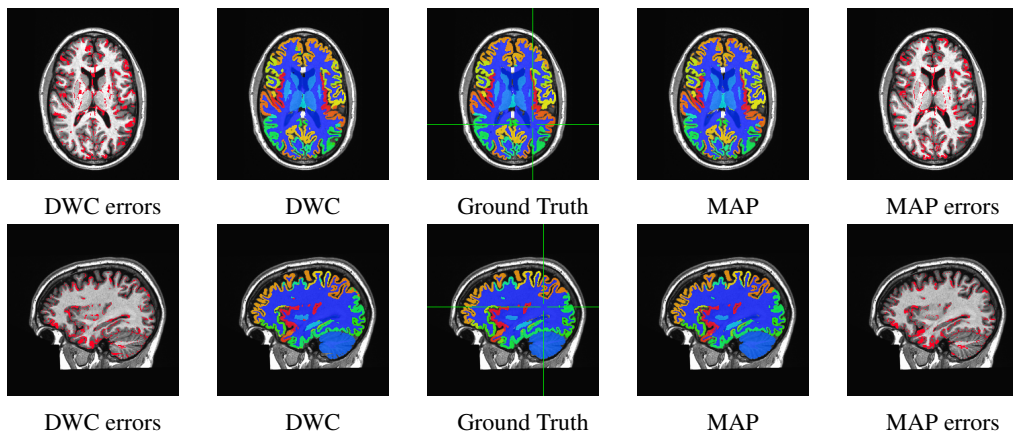

Figure 4: The axial and sagittal segmentations produced by DWC and the $HNBW_{MAP}$ baseline on a WU120 subject. The subject was selected by matching the subject specific Dice with the average Dice across WU120. Segmentations errors for all classes are shown in red in the respective plot.

led to ~3x, ~2x, and ~4x convergence speedups for HCP to NKI, HCP to Buckner and HCP to WU120, respectively. The batch-size was set to 10. Weight normalization [28] was used for the weight means for all networks and the weight standard deviations were initialized to 0.001 as in [19] for the variational network trained on HCP. For MAP networks and the variational network trained on HCP, $p(\mathbf{w}) = \mathcal{N}(0, 1)$.

### 3.1.3 Performance Metric

To measure the quality of the produced segmentations, we calculated the Dice coefficient, which is defined by

$$Dice_c = \frac{2|\hat{\mathbf{y}}_c \cdot \mathbf{y}_c|}{||\hat{\mathbf{y}}_c||^2 + ||\mathbf{y}_c||^2} = \frac{2TP_c}{2TP_c + FN_c + FP_c}, \tag{18}$$

where $\hat{\mathbf{y}}_c$ is the binary segmentation for class $c$ produced by a network, $\mathbf{y}_c$ is the ground truth produced by Freesurfer, $TP_c$ is the true positive rate for class $c$, $FN_c$ is the false negative rate for class $c$, and $FP_c$ is the false positive rate for class $c$. We calculate the Dice coefficient for each class $c$ and average across classes to compute the overall performance of a network.

### 3.1.4 Baselines

We trained MAP networks on the HCP ($H_{MAP}$), NKI ($N_{MAP}$), Buckner ($B_{MAP}$) and WU120 ($W_{MAP}$) datasets. We averaged the output probabilities of the $H_{MAP}$, $N_{MAP}$, $B_{MAP}$, and $W_{MAP}$ networks to create an ensemble baseline. We also trained a MAP model on the aggregated HCP, NKI, Buckner, and WU120 training data ($HNBW_{MAP}$) to estimate the performance ceiling of having the training data from all sites available together.

### 3.1.5 Variational Continual Learning

We trained an initial FFG variational network on HCP (H) using $H_{MAP}$ to initialize the network. We then used used VCL with HCP as the prior for distributed training of the FFG variational networks on the NKI ($H \rightarrow N$), Buckner ($H \rightarrow B$) and WU120 ($H \rightarrow W$) datasets. Additionally, we trained networks using VCL to transfer from HCP to NKI to Buckner to WU120 ($H \rightarrow N \rightarrow B \rightarrow W$) and from HCP to WU120 to Buckner to NKI ($H \rightarrow W \rightarrow B \rightarrow N$). These options test training on NKI, Buckner, and WU120 in increasing and decreasing order of dataset size, since dataset order may matter and may be difficult to control in practice.

### 3.1.6 Distributed Weight Consolidation

For DWC, our goal was to take distributed networks trained using VCL with an initial network as a prior, consolidate them per Eq. 17, and then use this consolidated model as a prior for fine-tuning on the original dataset. We used DWC to consolidate $H \rightarrow N$, $H \rightarrow B$, and $H \rightarrow W$ into $H \rightarrow N + B + W$ per Eq. 17. VCL [21] performance was found to be improved by using coresets [1, 12], a small sample of data from the different training sets. However, if examples cannot be collected from the different datasets, as may be the case when examples from the separate datasets cannot be shared, coresets are not applicable. For this reason, we used $H \rightarrow N + B + W$ as a prior for fine-tuning (FT) by training the network on H ($H \rightarrow N + B + W \rightarrow H$) and giving $\mathcal{L}_\mathcal{D}$ the weight of one example volume.

## 3.2 Experimental Results

In Table 2 we show the average Dice scores across classes and sMRI volumes for the differently trained networks. The weighted average Dice scores were computed across H, N, B, and W by weighing each of the Dice scores according to the number of volumes in each test set. For the variational networks, 10 MC samples were used during test time to approximate the expected network output. The weighted average Dice scores of DWC were better than the scores of the ensemble, the baseline method for combining methods across sites, ($p$ = 1.66e-15, per a two tailed paired $t$-test across volumes). The ABIDE Dice scores of DWC were not significantly different from the scores of the ensemble ($p$ = 0.733, per a two tailed paired $t$-test across volumes), showing that DWC does not reduce generalization performance for a very large and completely independent multi-site dataset.

| $Network$ | $H$ | $N$ | $B$ | $W$ | Avg. | $A$ |
|---|---|---|---|---|---|---|
| $H_{MAP}$ | 82.25 | 65.88 | 67.94 | 70.88 | 72.92 | 55.25 |
| $N_{MAP}$ | 71.20 | 72.19 | 70.73 | 73.06 | 71.66 | 66.67 |
| $B_{MAP}$ | 65.69 | 50.17 | 82.02 | 68.87 | 59.25 | 50.23 |
| $W_{MAP}$ | 70.18 | 66.27 | 72.20 | 76.38 | 68.76 | 62.83 |
| $H \rightarrow N$ | 75.40 | 73.24 | 71.77 | 73.17 | 74.03 | 64.62 |
| $H \rightarrow B$ | 73.85 | 56.79 | 79.49 | 68.53 | 65.78 | 49.27 |
| $H \rightarrow W$ | 77.07 | 67.63 | 76.15 | 77.26 | 72.51 | 62.31 |
| $H \rightarrow N \rightarrow B \rightarrow W$ | 77.42 | 71.46 | 79.70 | 79.82 | 74.86 | 63.3 |
| $H \rightarrow W \rightarrow B \rightarrow N$ | 78.04 | 78.15 | 75.79 | 79.50 | 77.99 | 70.79 |
| $H \rightarrow N + B + W$ (DWC w/o FT) | 78.28 | 73.52 | 78.02 | 77.37 | 75.95 | 65.56 |
| $Ensemble$ | 79.13 | 72.32 | 80.02 | 78.84 | 75.94 | 66.27 |
| $H \rightarrow N + B + W \rightarrow H$ (DWC) | 80.34 | 73.64 | 77.46 | 78.10 | 76.82 | 66.21 |
| $HNBW_{MAP}$ | 81.38 | 77.99 | 80.64 | 79.54 | 79.62 | 70.76 |

Table 2: The average Dice scores across test volumes for the trained networks on HCP (H), NKI (N), Buckner (B), and WU120 (W), along with the weighted average Dice scores across H, N, B, and W and the average Dice scores across volumes on the independent ABIDE (A) dataset.

Training on different datasets sequentially using VCL was very sensitive to dataset order, as seen by the difference in Dice scores when training on NKI, Buckner, and WU120 in order of decreasing and increasing dataset size ($H \rightarrow N \rightarrow B \rightarrow W$ and $H \rightarrow W \rightarrow B \rightarrow N$, respectively). The performance of DWC was within the range of VCL performance. The weighted average and ABIDE dice scores of DWC were better than the $H \rightarrow N \rightarrow B \rightarrow W$ Dice scores, but not better than the $H \rightarrow W \rightarrow B \rightarrow N$ Dice scores.

In Figures 1, 2, 3, and 4, we show selected example segmentations for DWC and $HNBW_{MAP}$, for volumes that have Dice scores similar to the average Dice score across the respective dataset. Visually, the DWC segmentation appears very similar to the ground truth. The errors made appear to occur mainly at region boundaries. Additionally, the DWC errors appear to be similar to the errors made by $HNBW_{MAP}$.

## 4    Discussion

There are many problems for which accumulating data into one accessible dataset for training can be difficult or impossible, such as for clinical data. It may, however, be feasible to share models derived from such data. A method often proposed for dealing with these independent datasets is continual learning, which trains on each of these datasets sequentially [4]. Several recent continual learning methods use previously trained networks as priors for networks trained on the next dataset [32, 21, 16], albeit with the requirement that training happens sequentially. We developed DWC by modifying these methods to allow for training networks on several new datasets in a distributed way. Using DWC, we consolidated the weights of the distributed neural networks to perform brain segmentation on data from different sites. The resulting weight distributions can then be used as a prior distribution for further training, either for the original site or for novel sites. Compared to an ensemble made from models trained on different sites, DWC increased performance on the held-out test sets from the sites used in training and led to similar ABIDE performance. This demonstrates the feasibility of DWC for combining the knowledge learned by networks trained on different datasets, without either training on the sites sequentially or ensembling many trained models. One important direction for future research is scaling DWC up to allow for consolidating many more separate, distributed networks and repeating this training and consolidation cycle several times. Another area of research is to investigate the use of alternative families of variational distributions within the framework of DWC. Our method has the potential to be applied to many other applications where it is necessary to train specialized networks for specific sites, informed by data from other sites, and where constraints on data sharing necessitate a distributed learning approach, such as disease diagnosis with clinical data.

**Acknowledgments**

This work was supported by the National Institute of Mental Health Intramural Research Program (ZIC-MH002968, ZIC-MH002960). JK's and SG's work was supported by NIH R01 EB020740.

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
