[Reviews · NeurIPS 2018]

Reviewer 1



The paper proposes a technique for learning a model by consolidating weights across models that are trained in different datasets. The proposed approach thus attempts to solve an important problem that arises by the limitations of sharing and pooling data. The authors take on the brain segmentation problem by using MeshNet architectures. The proposed method essentially starts from the model learned from one dataset, performs variational continual learning to parallel train across multiple datasets, and then performs bayesian parallel learning to fine tune the model on a dataset by using as prior the weights learned in parallel from the rest of the datasets. The proposed approach has been tested using free surfer segmentations for data part of the Human Connectome Project, the Nathan Kline Institute, the Buckner Lab and the ABIDE project. The proposed approach shows advantageous performance in a left out dataset when compared to simpler continual learning and MAP approaches. The paper is clear, well-written and organized. The figures are relatively small and difficult to appraise. Maybe a collage of the different panels would be more appropriate to present the results. There some author comments from the draft version that should be removed, and it is not clear what the Peking dataset is. I would also suggest the author tone down the “in parallel” argument given the sequential nature of the algorithm. Reading the paper, I was expecting to see a method that takes models that have been derived totally independently from each other and it optimally merges them, which is not the case. Additionally, and regarding the datasets, it would be useful to know what the strength of the scanners is, and if any additional preprocessing steps were taken to account for inhomogeneities, etc. The proposed approach is interesting, but it seems to be a creative combination of existing ideas to perform variational continual learning in parallel. Having said that, my most important concern regarding the paper is its experimental validation. First I find unfortunate the use of the segmentation task to exemplify its value. First, the use of free surfer segmentations as ground truth is problematic. Free surfer itself makes segmentation errors, and now we have a technique that makes even more errors. Second, sharing segmentation results and derivative datasets such as features extracted by ROIs is something that is commonly done, as exemplified by the success of the ENIGMA consortia. It is quite straightforward to run free surfer in every site and then pool meta-information to do analyses. Another task would be more appropriate. Importantly, I would have appreciated a more expensive validation. For example, how important is the choice of the starting dataset? How the results would change if NKI or Buckner data would have been used first? Critically, I would like to know what is the performance when training with all 3 datasets at the same time. This should be the ceiling that can be achieved. Last but not least, I find the ensemble baseline a bit poor. I would have preferred an ensemble method that learns how to combine the different models than simply average the probabilities.

Reviewer 2



Research is becoming an increasingly collaborative endeavor but experiments that result in data collection are still costly. The costs, together with privacy, regulations and practicality concerns lead to the current situation when large datasets are amassed across multiple research groups yet any single group rarely has enough data to train a machine learning algorithm. The paper proposes an algorithm for training machine learning models in these settings in the continual learning style, when a model is first trained on data at some of the sites and then the learned weights are aggregated and subsequent models are fine-tuned from that point. This is achieved thanks to a variational approximation and independence assumption on filter weights My two major concerns are 1. The unrealistic testing setup that goes against the offered decentralized research centers models. The test in the paper are conducted on a very few sites with a very large amount of data, and not on many sites with moderate data sizes. To gain an intuition whether the approach works one would need many more realistic setups for validation. 1. The choice of brain segmentation problem to demonstrate PWC. It is a well known feature of most of the neural segmentation methods that unlike other deep learning algorithms they only need a few labeled examples, since each pixel/voxel is labeled and provides a signal. Thus, among all of the deep learning algorithms the paper seems to imply it will improve the chosen method seems to be the one benefiting the least. Furthermore, it would be good to see and compare with performance of the original MeshNet segmentation algorithm just on the data that was used at the final stages. I suspect, it may perform as well. Other concerns: 1. line 52, in fact, MeshNet, when ran uses quite a lot of memory, may be even more than other mentioned methods, but that memory is in the activations, not in useful weights. So may be the paper needs to be more explicit of the fact, that MeshNet is the only one of the mentioned models that is feasible to use in the proposed settings as all other models require transfer of Gigabytes of parameters for each trained model, while MeshNet's parameters amount to under a Megabyte? 2. The cited papers of segmentation techniques for brain imaging are just the most recent. PyramidLSTM from 2015 is not cited, for example. 3. It is unclear from the text whether the algorithm ships the weights over the network or what happens to them at all. it would be good to have that explained. 4. Line 77-78 - editorial comment is left in the manuscript. 5. independent values of w in spatial filter are hard to assign the meaning to. The whole purpose of the local spatial filter is to capture correlations in the input, so the weights cannot be independent. 6. The notation is heavy and is not optimized for getting the point across. The use of indexing is excessive and leads to difficulties with interpreting even simple mathematics. Equation 16 is an example that could be greatly simplified but then it would took quite simple. 7. Line 141 implies that parameter uncertainty increases with each consolidation, If that is try then does not that defeat the point of the work? 8. The model uses a truncated set of FreeSurfer units, while the complete 104 ROI set in the atlas could be more useful for practicioners. 9. line 153 provides citations for splitting each volume, but the provided references did not split, but sample subvolumes. 10. Lines 158-159 needs to qantified. 11. lines 171-173 the vanilla MeshNet should be also among the baselines. 12. it may be worth citing prior work on successfully executing the model that the authors describe here (ENIGMA 1) and potentially any of the papers that worked out the decentralized data setup in detail, including a similar to the proposed "hot potato" algorithm schema as well as the iterative one (COINSTAC papers 2). Without those citations the current language in the paper misleads the reader to think that the specific decentralized biomedical setup is also proposed by the authors of this paper. 1. Thompson, P. M. and others (2014). The ENIGMA consortium 2. Plis, SM et al., 2016. COINSTAC: a privacy enabled model 1. lines 43-45 imply that the datasets were combined but line 153 states otherwise 2. Lines 199-200 The vision does come across clear. 3. line 203 - unsupported claim. 4. Why the subvolumes size have been decreased from 38 to 32? 5. The receptive field size of convolutional network is 37, which is bigger then 32 subvolume size Minor: - line 94 - such elastic, should be such as elastic - line 95 - reinterpretted - an extra t

Reviewer 3



This is an interesting practical idea. The difficulty of sharing health-related datasets is a real challenge and sharing trained deep learning models would be practically useful. This work provides a hypothetical situation of multiple sites collecting the same data that can be pooled into training a single deep learning model. The need for a method such as Parallel Weight Consolidation (PWC) is 1) so that trained models are passed around (not the data) between sites, 2) subsequent learning is faster, and 3) computational resource requirements are lower. Strengths: simple and straightforward way to combine trained deep learning model weights. Weaknesses: with any practical idea, the devil is in the details and execution. I feel that this paper has room for improvement to more thoroughly compare various choices or explain the choices better (e.g. choice of MeshNet and the underlying setting of multiple sites sharing the same exact deep learning model, etc. See below) Here is what I mean: 1. There is no comparison of using different network architectures. MeshNet may be just fine; however, it is not clear if other base learning methods can compare because it is impractical to assume MeshNet to represent a typical performance outcome 2. There is no baseline comparison: i.e. something like (H+N+B)_MAP that the last row of Table 2 should be compared to. Without a baseline comparison, it is not clear how much is lost with making the practical tradeoffs by using PWC. 3. Subsequent learning procedure is stated to be "much faster," (lines 158) but no formal comparison is given. I feel that this is an important comparison since the paper claims this is a difficulty individual sites have (lines 33-36) 4. What are the computational resources used to fit (aside from time) that makes continual learning easier? 5. Table 2 seems reasonable; however, I have questions about comparing apples to apples in the sense that comparisons should be made between using the same amount of data. For example, H->N and H->B use less data than H->N+B. Also, H->N->H and H->N->H use less data than H->N+B->H.